# Maternal emotional and physical intimate partner violence and early child development: investigating mediators in a cross-sectional study in a South African birth cohort

Whitney Barnett ![ORCID],[1] Sarah L Halligan,[2,3] Catherine Wedderburn,[1,4] Rae MacGinty,[1] Nadia Hoffman,[3] Heather J Zar,[1] Dan Stein,[3,5] Kirsten Donald ![ORCID] [5,6]

For numbered affiliations see end of article.

**Correspondence to**
Dr Whitney Barnett;
barnett.whitney@gmail.com

## ABSTRACT

**Objectives** This study investigated associations between recent maternal intimate partner violence (IPV) (emotional, physical and sexual) and child development at 2 years as well as whether maternal depression or alcohol use mediated these relationships.

**Design** Cross-sectional study nested in a South African birth cohort.

**Setting** Two primary care clinics in Paarl, South Africa.

**Participants** 626 mother–child pairs; inclusion criteria for maternal antenatal enrolment were clinic attendance and remaining in the study area for at least 1 year; women were excluded if a minor.

**Primary outcome measures** Child cognitive, language and motor development composite scores. These were assessed using the Bayley Scales of Infant and Toddler Development, third edition.

**Results** Emotional IPV was associated with lower cognitive (β=−0.32; 95% CI −0.60 to −0.04), language (β=−0.36; 95% CI −0.69 to −0.01) or motor composite scores (β=−0.58; 95% CI −0.95 to −0.20) in children at 2 years of age. Physical IPV was associated with lower motor scores (β=−0.42; 95% CI −0.75 to −0.09) at 2 years. Sexual IPV was unrelated to developmental outcomes, possibly due to low prevalence. Neither recent maternal depression nor alcohol use were shown to mediate the relationship between IPV and developmental outcomes.

**Conclusions** Interventions to reduce maternal physical and emotional IPV and early-life interventions for infants and toddlers are needed to promote optimal child development.

## BACKGROUND

Intimate partner violence (IPV) is one of the most common forms of violence against women. The WHO's multicountry study on violence against women found high lifetime prevalence of physical IPV (up to 61%), sexual IPV (up to 59%) and emotional IPV (up to 75%).[1] The epidemic of IPV is particularly marked in low-income andmiddle-income countries (LMICs), including

### Strengths and limitations of this study

▶ This study investigates the unique impact of multiple types of intimate partner violence (IPV) (emotional, physical and sexual) on developmental outcomes in very young children in a high-risk setting.

▶ A key strength of this study includes measurement of a broad range of risk factors and robust measurement of developmental outcomes in a representative sample of participants in a low-income and middle-income country setting.

▶ Understanding the factors underlying associations between IPV and poor child development is critical to better inform interventions and policy, thus, we investigated both maternal postnatal depression and alcohol use as potential mediators of IPV–child development relationships in this cohort.

▶ The current study used cross-sectional data and therefore did not investigate longitudinal exposures and cannot assert causality.

South Africa, where prevalence is among the highest globally.[2 3] Children in households with IPV are often witnesses and are at increased risk for abuse or neglect.[4 5] The United Nations estimates that each year between 133 and 275 million children witness violence between primary caregivers.[6] Exposure to violence in the home is linked to many adverse health outcomes in children, including developmental, psychological and behavioural impairments.[7–9] Importantly, these often impact health and development beyond childhood, having negative effects that extend into adolescence and adulthood.[8 10] However, there remain several gaps in the existing literature.

Relatively little research has examined associations between the presence of IPV and poorer cognitive, language and motor

development in infancy and very early childhood[11–13] though it is a critical developmental period that can have significant implications for long-term health.[14] Where data are available, exposure to IPV in the first 2 years of life has been associated with increased risk of developmental delay.[12 13] Healthy development during the first years of life is vital, given its impact on academic performance and mental health across the lifespan.[10 14] Indeed, a longitudinal study through 8 years found that exposure to interpersonal trauma (maltreatment and/or witnessing IPV) in the first 2 years of life was associated with the poorest cognitive outcomes compared with other periods in childhood.[15]

Research focusing on child development and IPV often narrowly defines IPV as only physical or sexual abuse,[12 13 16] though emotional IPV has been linked to similar or increased rates of ill-health, both physical and mental, compared with physical IPV.[17–20] Additionally, the majority of studies, particularly in school-aged children, have been done in high-income countries,[8 11 21] though LMICs have a higher burden of IPV-associated risk factors, such as community violence, trauma, substance use disorders and mental illness and limited support networks. South Africa in particular has among the highest burdens of interpersonal violence in the world.[22] The roots of high levels of violence can be linked to South Africa's colonial and apartheid past, including the impacts of systemic racism, impoverishment, migrant labour, gender norms and constructions of masculinity; their ongoing legacy continues to impact extremely high levels of violence in South Africa.[23] Zembe et al investigated social risk factors and relationship power inequity and IPV in a South African sample, finding high acceptability of violence in intimate relationships as well as gender norms supporting aggressive masculinity and subservient femininity, particularly in historically marginalised groups such as those included in this study.[24]

Lastly, a key gap in the existing literature examining associations between IPV exposure and early life child developmental outcomes is investigation of potential mediators in this relationship, particularly in LMIC settings. Several potential mechanisms underlying the relationship between IPV and child development have been proposed. For instance, maternal depression as well as alcohol use have been separately linked to both IPV and development outcomes,[13 20] potentially compromising maternal–child attachment, impacting maternal autonomy and caregiving, diverting financial resources in the home and may increase the risk of child maltreatment or abuse.[25] Alcohol is one of the most commonly misused substances, with extremely high prevalence of alcohol use disorders in South Africa,[26] and maternal use postnatally is associated with poorer child development and academic performance.[27–29] Maternal depression has been linked to insecure mother–child attachment[30]; less sensitive and responsive caregiving,[31] and reduced healthcare service utilisation[32] adversely impacting care and stimulation needed for healthy development. Although

there is evidence linking proposed mediators to both IPV and early-life development, extremely few have formally investigated potential explanatory factors.[33]

This study aims to explore associations between IPV subtypes (emotional, physical and sexual) and child development (cognitive, language and motor) at 2 years of age and to investigate potential mechanisms underlying this relationship. The current study addresses three key gaps in the literature, specifically by (1) investigating associations between IPV and development in very young children (2) investigating the differential impact of subtypes of IPV and (3) investigating maternal depression or alcohol use as potential mediators in these relationships. Given the high prevalence of IPV in low resource settings, understanding the significant impact IPV exposure may have on young children during this developmentally sensitive period and improved understanding of possible pathways underlying these relationships is important for improving child potential globally.[11]

## METHODS

The parent study is a longitudinal, population-based birth cohort study following mother–child dyads from pregnancy through childhood.[34–36] The study is located in a periurban area in South Africa, characterised by low socioeconomic status and psychosocial risk factors. There is a free primary healthcare system, that includes antenatal and child health services. The current cross sectional analysis uses data from three study visits occurring at or near 2 years of child age; only birth weight and HIV exposure were collected prior to 2 years of age, at birth, as described below.

### Participants

Pregnant women were enrolled from March 2012 to March 2015. Women were enrolled in their second trimester, between 20 and 28 weeks' gestation at two public sector primary healthcare clinics, one serving a predominantly mixed-ancestry population and the other serving a predominantly black African population. Our intention in using the historic terms of black African and mixed ancestry is not meant to reify these terms, but to contribute to the literature on ongoing health disparities. Inclusion criteria were: (1) pregnant women attending one of the two study clinics; (2) aged 18 years or older and (3) intending to remain in the study area for at least 1 year.

Mothers gave written informed consent at enrolment and were reconsented annually thereafter. Study staff were trained in ethical conduct of violence and mental health research, including confidentiality and identification and management of risk. Interviews were conducted privately, data were deidentified and only accessible to study staff to ensure confidentiality. Where significant mental health issues or safety concerns were identified, staff referred participants to appropriate care or social

services (including support services for IPV, substance abuse and mental health issues).

## Clinical and sociodemographic data

Sociodemographic information was collected from the mother at a visit 2 years after their child's birth, using a standardised questionnaire adapted from items included in the South African Stress and Health Study.[37] Maternal education (any secondary vs completed secondary), employment and partnership status (single or married/marriage-like relationship) and household income (<R1000/month (US$60), R1000–R5000 (US$350) or >R5000/month) were self-reported at 2 years postpartum. Maternal HIV status during pregnancy was confirmed by routine testing during antenatal care. All HIV-exposed children were tested per the Western Cape prevention of mother-to-child transmission guidelines.[38] Birth weight and postnatal anthropometric measurements were collected by trained staff at birth and 2 years, as has been described.[39]

## Intimate partner violence

Recent maternal IPV was measured at 2 years post partum using the Intimate Partner Violence Questionnaire (IPVQ), a 12-item inventory adapted from the WHO multicountry study[1] and the Women's Health Study in Zimbabwe.[40] The IPVQ assessed recent (past-year) exposure to emotional (4 of 12 questionnaire items), physical (5 of 12 items) and sexual abuse (3 of 12 items). Partner behaviour indicating emotional IPV included having been insulted or made to feel bad, having been humiliated in front of others, intentionally scared or intimidated or threatened with physical harm. Physical IPV included being slapped, pushed, shoved, hit with an object, beaten or choked. Sexual IPV exposure was classified based on having been forced to have sex, afraid not to have sex or forced to do something sexual which was degrading or humiliating. Mothers were asked about frequency of exposure to partner behaviour ('never', 'once', 'a few times' or 'many times'). Items were then summed to create scores for each IPV subtype with higher scores indicating greater severity and frequency of exposure; exposure was defined as a score >1 indicating more than an isolated incident within each subtype. Scoring guidelines were devised for the purposes of this study, and were based on prior work in South Africa.[41]

## Proposed mediating variables

Recent maternal depression was measured using the Beck Depression Inventory (BDI), a validated and reliable screening tool for depressive symptoms at 2 years post partum.[42 43] The BDI-II comprises 21 items, which assesses severity of symptoms of major depression on a severity scale from 0 (absence of symptoms) to 3 (severe). Individual items are summed to obtain a total score, with higher scores indicating more severe depressive symptoms. A cut-off score of ≥20 was used to indicate participants with probable depression.[44] The Alcohol, Smoking and Substance Involvement Screen Test (ASSIST) was used to measure current maternal alcohol-related risk at 2 years of child age.[45] Alcohol-related items were summed to create a score for alcohol-related problems with higher scores indicating greater substance-related risk. We used published guidelines for the ASSIST to classify alcohol scores into three categories for descriptive purposes: low/no risk (0–10), moderate risk (11–26) and high risk (>26), with the likelihood of alcohol dependence.[46]

## Bayley scales of infant and toddler development, third edition

The Bayley Scales of Infant and Toddler Development, third edition (BSID-III)[47] was used to measure toddler neurodevelopment. It is widely used to assess development in children ages 2–42 months and has been validated for use in South Africa.[48] Assessments for neurodevelopment were done at 2 years of child age by trained physiotherapists and occupational therapists supervised by a paediatric neurodevelopmental specialist.[36] Quality control and monitoring processes were implemented to ensure accuracy. Assessments were conducted with language prompts in the child's preferred language and assessors alternated between sites and assessed a similar number of children. In the current study, composite scores for cognition, language and motor development were used. The composite scores are calculated using a normal US population, and are standardised to have a mean of 100 and SD of 15. Composite scores are strongly correlated to raw scores in the current study and allow comparison across domains. Continuous composite scores as well as categorical outcomes are reported. Categories were defined as 'delay' (<2 SD below mean), 'suboptimal development' (<1 SD below mean) and 'no delay'.

## Patient and public involvement

Investigators have established close relationships with key stakeholders including Western Cape Government Health Department and community members. Participants were not involved in the design or recruitment of the Drakenstein Child Health Study (DCHS). However, participant experience and satisfaction have been assessed at multiple time points.[49] Participant feedback about study experience, information provided and study procedures have been used ongoing to reassess study protocols and to make alternations to improve study experience and participant satisfaction. Study findings are routinely fed back to the community and the Department of Health clinical and administrative staff.

## Data analysis

The analyses were conducted using STATA V.15 (StataCorp). Demographic, clinical, psychosocial and child development data were described using median (IQR) for continuous data or number (%) for categorical data. Developmental data were compared by child sex using Pearson's $\chi^2$ test for categorical data and the Wilcoxon rank-sum test for continuous data.

Mean composite scores were calculated for the Bayley-III for cognitive, language and motor domains. Birth weight was converted to z-scores based on child sex and gestational age using the INTERGROWTH-21st standards.[50] Postnatal weight-for-age z-scores, length-for-age z-scores and weight-for-length z-scores were calculated using weight and length measurements at 24 months, using Anthro software (WHO, Geneva, Switzerland).[51] These as well as emotional, physical and sexual IPV and proposed mediator variables were included as continuous scores in all analyses.

We examined associations using a complete case analysis sample. Bivariate linear regression analyses were used to investigate the relationship between developmental domains, clinical and demographic variables, proposed mediators and IPV subtypes. Regression coefficients with 95% CIs are reported. Multivariable linear regression models were then run for each of the developmental (composite cognitive, language and motor scores) outcomes, including significant univariable coefficients (p<0.05). Continuous scores were used for exposure, mediator and outcome data in multivariable models. Where multiple IPV subtypes were significant in bivariate regression, adjusted models were run for each separately, as these were highly correlated. Hypothesised mediators of the relationship between IPV and early life development (current maternal depression and alcohol use) were investigated. Criteria for formal investigation of potential mediators were (1) their association with developmental outcomes, using a cut-off level of p<0.05, in final adjusted models (path b) and (2) where associations between IPV and the hypothesised mediator were significant (path a).

In addition to the analysis of observations with complete data, we examined associations after imputing missing data for covariates and developmental outcomes using multiple imputation with chained equations. We assumed covariate data were missing at random and performed 30 imputations. Results from complete case and imputed datasets produced similar conclusions. Results based on imputed data are presented in online supplemental tables. To explore potential selection bias, we assessed the characteristics of mother–child dyads with complete and missing data, using Pearson's $\chi^2$ and Wilcoxon rank-sum tests to compare covariate distributions between groups.

## RESULTS

Of 1143 study births, 154 (13.5%) children were lost to follow-up before reaching 2 years of age primarily due to moving out of the study area or being uncontactable or unavailable for follow-up visits. A further 255 (25.8%) did not complete a BSID-III assessment due to non-attendance on the day or moving out of the study area. Data for the current study were collected across three visits at approximately 2 years of child age: a clinical visit for growth and sociodemographic data collection; a psychosocial visit for maternal depression and substance use data and a visit assessing child development. A total of 626 children who completed the BSID-III assessment and had IPV exposure and covariate data were included in the final complete case dataset. Comparing the characteristics of dyads with complete and missing exposure or outcome data, mothers with missing data were less likely to be HIV infected (p=0.01) and less likely to be employed (p=0.02, online supplemental table 1).

Clinical, demographic and psychosocial characteristics at 2 years are presented in table 1. This sample was characterised by low maternal employment (41%), low maternal education (37% completed secondary education), low household income (only 27% earning more than R5000 per month (approximately US$350)). Almost half of mothers were married or cohabiting (49%) and 24% of children were HIV exposed with only two children HIV infected. A minority of mothers experienced psychosocial risk factors or substance use, including emotional IPV (16%), physical IPV (17%) and sexual IPV (4%), alcohol (10% moderate/high risk) and depression (6%) (table 1).

### Developmental outcomes

A total of 28/625 (4%) children were categorised as having cognitive delay (<2 SD below mean), 48/598 (8%) with language delay and 7/592 (1%) with motor delay. Only language differed significantly by child sex, with males having a higher prevalence of delay compared with females (10% vs 5%; p=0.001) and lower composite scores (p<0.001) (table 2).

### IPV and developmental outcomes

At 24 months, bivariate analysis showed that emotional IPV was associated with lower composite cognitive scores (β=−0.32; 95% CI −0.58 to −0.06), lower composite language scores (β=−0.35; 95% CI −0.69 to −0.01) and lower composite motor scores (β=−0.58; 95% CI −0.93 to −0.24). Physical IPV was associated with lower composite motor scores (β=−0.42; 95% CI −0.73 to −0.11). Sexual IPV was not associated with any composite developmental outcomes (table 3).

Multivariable regression models were run to examine associations between IPV and child developmental outcomes, adjusting for key potential confounding factors. IPV subtypes were run separately in adjusted models due to collinearity and only where significant in bivariate analyses. In adjusted models, the results held, recent emotional IPV was associated with reduced scores across all developmental outcomes, specifically with lower composite cognitive scores (β −0.32; 95% CI −0.60 to −0.04), language (β −0.35; 95% CI −0.69 to −0.01) and motor scores (β −0.58; 95% CI −0.95 to −0.20). Physical IPV was associated with lower motor scores (β −0.42; 95% CI −0.75 to −0.09) (table 4). Associations using the imputed dataset were similar for key exposures (IPV) and outcomes (developmental) (online supplemental tables 2 and 3).

### Hypothesised mediators

Current maternal depression and alcohol use were explored as potential mediators in the relationship

**Table 1** Sample maternal and child psychosocial, demographic and clinical characteristics, n=626*

|  | Total |
|---|---|
|  | n=626 |
| **Intimate partner violence (24 months)** | |
| Emotional score, median (IQR) | 0 (0, 0) |
| Emotional (above threshold), n (%) | 102 (16) |
| Physical score, median (IQR) | 0 (0, 0) |
| Physical (above threshold), n (%) | 107 (17) |
| Sexual score, median (IQR) | 0 (0, 0) |
| Sexual (above threshold), n (%) | 25 (4) |
| Any subtype score, median (IQR) | 0 (0,0) |
| Any subtype (above threshold), n (%) | 138 (22) |
| **Sociodemographics (24 months)** | |
| Maternal education (secondary not completed), n (%) | 233 (37) |
| Mother employed, n (%) | 225/545 (41) |
| Mother married/partnered, n (%) | 268/544 (49) |
| **Household Income** | |
| <R5000 (approximately 300 USD), n (%) | 445/609 (73) |
| >R5000, n (%) | 164/609 (27) |
| Recruitment site: TC Newman, n (%) | 274 (44) |
| **Physical variables** | |
| Birth weight z-score, median (IQR)† | −0.59 (−1.34, 0.16) |
| Child age at assessment (months), median (IQR) | 24.05 (23.79, 24.31) |
| HIV exposed†, n (%) | 153 (24) |
| Length for age z-score at 24 months, median (IQR) | −1.16 (−1.89 to 0.52) |
| Weight for age z-score at 24 months, median (IQR) | −0.31 (−1.10, 0.43) |
| Weight for length z-score at 24 months, median (IQR) | 0.28 (−0.48, 1.21) |
| **Psychosocial variables (24 months)** | |
| Depression score, median (IQR) | 0 (0, 3) |
| Depression (above threshold), n (%) | 36/591 (6) |
| Alcohol dependence score, median (IQR) | 0 (0, 0) |
| **Alcohol dependence (categories)** | |
| Low dependence, n (%) | 160/593 (27) |
| Moderate dependence, n (%) | 33/593 (6) |
| High dependence, n (%) | 22/593 (4) |

Median (IQR) ranges reported for anthropometry data, intimate partner violence scores as well as depression and alcohol scores. Frequencies (percentages) reported for all other variables. All percentages calculated on non-missing values. N values are indicated where the number of participants with available data differs from the total group (n=626). Intimate partner violence, above threshold exposure defined as more than an isolated incident within each subtype. Depression threshold defined as BDI score of ≥20. Alcohol dependence categories defined as: low/no risk (0–10), moderate risk (11–26) and high risk (>26), with the likelihood of alcohol dependence.
*Birth weight and HIV exposure collected at birth; all other variables collected at 2 years of child age.
†Includes data from mothers or children included in complete case dataset.
BDI, Beck Depression Inventory.

between recent IPV subtypes and developmental outcomes. In correlational analyses, both maternal depression and alcohol use were positively associated with emotional as well as physical IPV (online supplemental table 4). However, when linear regressions were run investigating associations between mediators and developmental outcomes, adjusted for potential confounding factors, we found no evidence of associations between proposed mediators and each developmental outcome (cognitive, language, motor), online supplemental table 5). Therefore, formal mediation analyses were not done.

**Table 2** Developmental outcomes of children at 2 years of age by child sex

|  | Total | Female | Male | P value |
|---|---|---|---|---|
| **Sample** | **n=625** | **n=303** | **n=322** |  |
| Cognitive scores, mean (SD) | 85.74 (8.84) | 86.44 (8.56) | 85.08 (9.06) | 0.035 |
| Cognitive categories |  |  |  |  |
| No delay, n (%) | 424 (68) | 218 (72) | 206 (64) | 0.078 |
| Suboptimal development, n (%) | 173 (28) | 75 (25) | 98 (30) |  |
| Delayed, n (%) | 28 (4) | 10 (3) | 18 (6) |  |
| Sample | n=598 | n=289 | n=309 |  |
| Language scores, mean (SD) | 84.45 (11.39) | 86.57 (11.35) | 82.47 (11.08) | <0.001 |
| Language categories |  |  |  |  |
| No delay, n (%) | 297 (50) | 164 (57) | 133 (43) | 0.001 |
| Suboptimal development, n (%) | 253 (42) | 110 (38) | 143 (46) |  |
| Delayed, n (%) | 48 (8) | 15 (5) | 33 (11) |  |
| Sample | n=592 | n=289 | n=303 |  |
| Motor scores, mean (SD) | 93.62 (11.89) | 94.40 (12.12) | 92.86 (11.63) | 0.111 |
| Motor categories |  |  |  |  |
| No delay, n (%) | 490 (83) | 243 (84) | 247 (82) | 0.251 |
| Suboptimal development, n (%) | 95 (16) | 41 (14) | 54 (18) |  |
| Delayed, n (%) | 7 (1) | 5 (2) | 2 (1) |  |

Continuous composite scores are presented as mean (SD). Categorical data presented as frequencies (percentages). Categories were defined as 'delay' (<2 SD below mean), 'suboptimal development' (<1 SD below mean) and 'no delay'. P values presented are comparing sex differences.

## DISCUSSION

Our results show that recent maternal emotional IPV was associated with lower cognitive, language and motor scores and physical IPV with lower motor scores in children at 24 months. Further, we investigated the potential role of current maternal depression or alcohol use in this relationship. There was insufficient evidence that current maternal depression or alcohol use were explanatory factors in the relationship between IPV and child development at 2 years of age.

We found associations between emotional IPV and each developmental outcome investigated, which are novel findings in such young children. Though there is limited research in this age group, and existing studies use differing definitions of IPV, a recent study pooling data from 11 LMICs investigating preschool development (36–59 months) found associations between all IPV subtypes and developmental outcomes.[52] Where investigated in a younger sample, studies have found a link between IPV and early life development. However, few investigated IPV subtypes as separate risk factors for poor development. For example, a US study, investigating the impact of IPV on developmental outcomes at 2 years reported an increased risk of language and neurological delay in infants and toddlers, however this was not analysed by IPV subtype but rather a composite measure of emotional, physical and sexual IPV.[12] A Tanzanian study through 3 years of age found both maternal lifetime physical and lifetime sexual IPV to be associated with reduced

motor skills, expressive and receptive communication and cognitive development.[13] Emotional IPV was not included in their IPV measure. One study that did investigate emotional IPV separately as a risk factor for cognitive and language delay, only found associations with postnatal physical IPV and only in unadjusted analyses. The limited sample size (n=72) may have reduced statistical power to detect an association.[53] In the current study, sexual IPV was not associated in unadjusted or adjusted analyses with the outcomes investigated, but this is likely due to insufficient power to detect an association given the relatively low prevalence of recent sexual IPV in our study population (4%). In the current study, maternal emotional IPV as compared with physical IPV was more consistently associated with poorer developmental outcomes. These results indicate that emotional IPV may carry a unique risk for poor child developmental outcomes in early life in our setting.

Given the high prevalence of emotional IPV and that it is often not as readily recognised or included in research, intervention or policy efforts aimed at prevention of IPV, it is critical to better understand its distinct impact on child health outcomes. Our findings suggest that adverse outcomes and their association with maternal IPV may be underestimated when IPV exposure is not considered broadly and inclusive of emotional IPV, a potentially significant gap in the literature. Emotional abuse in a current or past intimate relationships is increasingly being recognised as a unique contributing factor to maternal

**Table 3** Univariate associations between intimate partner violence (IPV), covariates and proposed mediators and composite scores for developmental domains in complete case analysis

|  | Composite cognitive (n=549) | Composite language (n=598) | Composite motor (n=521) |
|---|---|---|---|
|  | Unadjusted | Unadjusted | Unadjusted |
|  | Coefficient (95% CI) | Coefficient (95% CI) | Coefficient (95% CI) |
| **IPV** | | | |
| Emotional IPV score | −0.32 (−0.58 to 0.06)* | −0.35 (−0.69 to 0.01)* | −0.58 (−0.93 to 0.24)* |
| Physical IPV score | −0.18 (−0.42 to 0.05) | −0.23 (−0.54 to 0.08) | −0.42 (−0.73 to 0.11)* |
| Sexual IPV score | −0.15 (−0.97 to 0.67) | −0.16 (−1.29 to 0.97) | −0.49 (−1.63 to 0.65) |
| **Sociodemographics** | | | |
| Maternal education (not completed secondary) | −1.31 (−2.70 to 0.08) | −2.86 (−4.61 to 1.11)* | −0.93 (−2.78 to 0.92) |
| Mother employed | 0.60 (−0.87 to 2.08) | 0.34 (−1.57 to 2.27) | 0.68 (−1.29 to 2.65) |
| Mother married/partnered | −0.52 (−1.98 to 0.93) | 0.47 (−1.43 to 2.37) | 0.01 (−1.94 to 1.95) |
| **Household income** | | | |
| >R5000 | 0.73 (−0.92 to 2.38) | 0.32 (−1.80 to 2.45) | 1.63 (−0.55 to 3.82) |
| Recruitment site: TC Newman | 1.19 (−0.16 to 2.53) | 3.05 (1.35 to 4.76)** | 1.13 (−0.66 to 2.93) |
| **Physical variables** | | | |
| Birth weight z-score | 0.42 (−0.17 to 1.01) | 0.84 (0.09 to 1.58)* | 0.62 (−0.17 to 1.39) |
| Male | −1.66 (−3.01 to 0.32)* | −4.27 (−5.96 to 2.59)** | −1.52 (−3.31 to 0.27) |
| Child age (months) | −0.94 (−2.23 to 0.34) | −0.10 (−1.74 to 1.53) | −2.81 (−4.49 to 1.12)* |
| HIV exposed | −1.71 (−3.30 to 0.11)* | −3.90 (−5.93 to 1.87)** | −0.76 (−2.89 to 1.38) |
| Length for age z-score | 0.97 (0.36 to 1.58)* | 0.77 (−0.01 to 1.55) | 0.13 (−0.70 to 0.97) |
| Weight for age z-score | 0.92 (0.32 to 1.51)* | 0.64 (−0.12 to 1.40) | 0.68 (−0.19 to 1.47) |
| Weight for length z-score | 0.61 (0.02 to 1.19)* | −0.34 (−0.41 to 1.09) | 0.84 (0.05 to 1.63)* |
| **Proposed psychosocial mediators** | | | |
| Depression score | −0.02 (−0.11 to 0.08) | −0.06 (−0.18 to 0.06) | −0.12 (−0.25 to 0.01) |
| Alcohol dependence score | −0.07 (−0.17 to 0.03) | −0.03 (−0.16 to 0.11) | 0.01 (−0.13 to 0.15) |

All variables measured at 2 years of child age except birth weight, sex and HIV-exposure which were captured at birth. IPV subtypes, depression and alcohol dependence were included as continuous variables in all models.
*p<0.05, **p<0.01.

**Table 4** Multivariable linear regression demonstrating the adjusted association of intimate partner violence (IPV) and composite scores for developmental domains in complete case analysis sample

|  | Composite cognitive (n=549) | Composite language (n=598) | Composite motor (n=521) | |
|---|---|---|---|---|
|  | Adjusted* | Adjusted† | Adjusted‡ | Adjusted‡ |
|  | Coefficient (95% CI) | Coefficient (95% CI) | Coefficient (95% CI) | Coefficient (95% CI) |
| Emotional IPV score | −0.32 (−0.60 to 0.04)* | −0.35 (−0.69 to 0.01)* | −0.58 (−0.95 to 0.20)* | |
| Physical IPV score | | | | −0.42 (−0.75 to 0.09)* |

Adjusted models were run only where IPV subtype was associated with outcome explored (p<0.05) in bivariate analyses. For cognitive and language scores, only emotional IPV was associated in bivariate analyses and therefore run in multivariable models. Due to collinearity, emotional and physical IPV were run in separate multivariable models for motor development. IPV subtypes, depression and alcohol dependence were included as continuous variables in all models.
*Adjusted for child sex, HIV exposure, length-for-age z-scores, weight-for-age z-scores and weight-for-length z-scores at 24 months.
†Adjusted for maternal education, recruitment site, weight-for-age z-score at birth, child sex and HIV exposure.
‡Adjusted for child age at assessment and weight-for-length z-scores at 24 months.

ill-health outcomes.[18–20] Specifically, emotional IPV has been linked to poor mental health, post-traumatic stress disorder and poorer physical health, sometimes above that of physical IPV.[19 54] Psychological control or abuse by a partner may uniquely impact maternal efficacy, independence and diminish maternal power within the family, which may affect her ability to provide nutritious food, a secure environment or to have positive parenting practices, all of which could lead to increased risk of negative outcomes for her children. These factors may be particularly influential for health outcomes in settings such as South Africa where entrenched gender norms and a high acceptability of violence contribute to high levels of IPV.[24]

Our investigation of potential mediators found little evidence to support that current maternal depression symptoms or reported alcohol use were explanatory factors for the impact of IPV on child development at 2 years. In the parent study, we have previously found that IPV is linked to longitudinal maternal depression[20] as well as hazardous alcohol use in pregnancy.[55] In this study, we found that IPV was associated with depression and alcohol use at 2 years. However, these did not appear to be associated with development at 2 years. Recent maternal depression or alcohol use may compromise the quality of parenting or parent–child attachment, may divert household financial resources and decrease maternal care-seeking behaviours, which in turn may compromise child development. Though there is much literature linking maternal substance use to adverse child outcomes, some studies that have investigated its impact on parenting or secure attachment have found mixed results.[56 57] For example, a study assessing maternal substance use and quality of caregiving found no differences between substance abusing and non-substance abusing mothers and caregiving or maternal–child attachment at 12 months.[56] Similarly, studies investigating maternal depression and child development have shown mixed findings.[13 58 59] A Tanzanian study investigating depression and toddler development, found that depression was associated with receptive language and motor development but not cognitive or expressive language.[13] Another study found that maternal depression at 7 weeks postpartum impacted development at 3 months but this relationship attenuated by 6 months.[58] Our findings may be influenced by relatively low levels of probable clinical depression (11%) or moderate-high alcohol-related risk (10%), diluting possible associations. We found no evidence that recent maternal depression or alcohol use influenced the relationship between IPV and early life development; these findings may illustrate the complexity of the relationship between the factors explored and other adverse environmental risks which often co-occur and may be underlying these relationships.

The prospective nature of the parent study, including measurement of a broad range of risk factors and robust measurement of developmental outcomes in an LMIC setting, is a strength of this study. However, there are several limitations. The current study used cross-sectional data, therefore, we cannot assert causality. Maternal sexual IPV prevalence was very low in our sample (4%) and this likely affected power to discern potential associations. We also did not investigate longitudinal exposures or cumulative impact of maternal depression or alcohol use; these were only investigated as recent exposures. Further, although the BSID-III is a well-recognised tool, further standardisation in sub-Saharan African settings is required. For example, the categorisation of delay is based on scaled scores using normative US data that might not be generalisable to a South African population. However, our data from the parent study has shown that raw scores have similar patterns to the results based on delay categorisation. There were some differences in characteristics of those included vs excluded in this study, namely HIV exposure and maternal employment. Maternal employment was lower in the included subsample (41% vs 51%), however, household income was similar across groups. Furthermore, maternal employment was not associated with any outcomes investigated in bivariate analyses and adjustment for this did not alter any of the key associations. HIV exposure was higher in the included subsample (24% vs 18%), however, this would likely bolster potential associations. In addition, approximately one-third of children who were still active in the study at 2 years were not included in the analysis due to incomplete data. However, results were similar when associations were explored in the imputed dataset.

## CONCLUSIONS

In summary, our results highlight the need for programmes that aim to reduce IPV as well as early life prevention efforts to mitigate the impact of maternal IPV on children. Maternal IPV was a consistent predictor of poor child developmental outcomes; furthermore, emotional IPV emerged as a uniquely deleterious risk factor for child developmental outcomes. Further work is needed to understand the mechanisms underlying this relationship, exploring maternal depression and alcohol use in other samples as well as additional potential mechanisms. Given our findings, it is important for public health professionals and policy makers to ensure that maternal emotional IPV is included in training programmes for healthcare staff as well as in screening and referral efforts. Lastly, more research investigating emotional or psychological IPV and adverse health outcomes is needed. Critically, in our setting, the majority of pregnant and early postpartum women access healthcare, providing an important, potentially underused opportunity to intervene early to attempt to shield children from the negative effects of IPV.

**Author affiliations**
[1]Department of Paediatrics and Child Health & South African Medical Research Council Unit on Child & Adolescent Health, University of Cape Town, Faculty of Health Sciences, Cape Town, South Africa
[2]Department of Psychology, University of Bath, Bath, UK

[3]Department of Psychiatry and Mental Health & South African Medical Research Council Unit on Risk and Resilience in Mental Disorders, University of Cape Town Faculty of Health Sciences, Cape Town, South Africa

[4]Department of Clinical Research, London School of Hygiene & Tropical Medicine, London, UK

[5]Neuroscience Institute, University of Cape Town Faculty of Health Sciences, Cape Town, South Africa

[6]Division of Developmental Paediatrics, Department of Paediatrics and Child Health, University of Cape Town Faculty of Health Sciences, Cape Town, South Africa

**Acknowledgements** We thank the study staff in Paarl, the study data team and lab teams, the clinical and administrative staff of the Western Cape Government Health Department at Paarl Hospital and at the clinics for support of the study. We thank the families and children who participated in this study.

**Contributors** WB, KD and SLH conceptualised the study aims and methodology. WB with guidance from RM, conducted all analyses. HJZ is principal investigator of the parent study; DS and NH lead the psychosocial study aspects; CW and KD are coinvestigators and were responsible for implementation, training and quality control of developmental assessments. All authors provided intellectual input and reviewed and approved the final manuscript. WB is responsible for the overall content as the guarantor.

**Funding** The study was funded by the Bill and Melinda Gates Foundation (OPP 1017641). Additional support for HJZ and DS is from the SA MRC. Additional aspects of the work reported here are supported by an Academy of Medical Sciences Newton Advanced Fellowship (NAF002\1001), funded by the UK Government's Newton Fund, by the UK MRC (MR/T002816/1) and by the US Brain and Behaviour Foundation Independent Investigator grant (24467). WB is supported by the SAMRC, through its Division of Research Capacity Development under the Bongani Mayosi National Health Scholars programme. CW was supported by the Wellcome Trust through a Research Training Fellowship (203525/Z/16/Z).

**Competing interests** None declared.

**Patient consent for publication** Not applicable.

**Ethics approval** Ethical approval was obtained from the Faculty of Health Sciences Research Ethics Committee, University of Cape Town (401/2009) and the Western Cape Provincial Research committee.

**Provenance and peer review** Not commissioned; externally peer reviewed.

**Data availability statement** Data are available on reasonable request. Data may be obtained from a third party and are not publicly available. Collaborations for the analysis of data are welcome; the parent study has a large and active group of investigators and postgraduate students and many have successfully partnered with students or researchers from other institutions. Researchers who are interested in datasets or collaborations can find more information on our website [http://www.paediatrics.uct.ac.za/scah/dclhs].

**ORCID iDs**
Whitney Barnett http://orcid.org/0000-0001-5082-7864
Kirsten Donald http://orcid.org/0000-0002-0276-9660

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
