## [Reviewer comments · BMJ Open]

ARTICLE DETAILS

TITLE (PROVISIONAL)	Maternal emotional and physical intimate partner violence and early child development: investigating mediators in a cross sectional study in a South African birth cohort.
AUTHORS	Barnett, Whitney; Halligan, Sarah; Wedderburn, Catherine; MacGinty, Rae; Hoffman, Nadia; Zar, Heather; Stein, Dan; Donald, Kirsten

VERSION 1 – REVIEW

REVIEWER	Peter Goldblatt University College London, UCL Institute of Health equity
REVIEW RETURNED	19-Jan-2021

GENERAL COMMENTS	This is a very well analysed set of data on IPV among mothers of two-year old children. The relationships between reduced child development and both emotional and physical IPV emotional are clearly set out and makes the paper well worth publishing. There are a few places in the paper where the description could be made clearer. In order of occurrence in the paper, these are as follows: Abstract Page 4, line 33 The statement "Interventions to reduce maternal IPV, including emotional IPV,... are needed" does not accurately reflect the findings in these populations as the research results focus only on Emotional and physical IPV. Suggest splitting this into a motherhood and apple pie statement about the desirability of intervening to reduce all forms of IPV and then highlighting that emotional and physical IPV interventions are needed in the populations to protect child development. Page 6, line 22 It would be helpful to clarify that the roots of violence are as much affected by the ongoing impact of past colonialism and apartheid on society - not just what they did to society in the past. Page 7, line 57 It would be helpful to describe here more explicitly what information was obtained longitudinally i.e. during pregnancy and at birth of the child and what was obtained cross sectionally i.e. at two years since the birth.
---

	Page 25, table 1 A range of different types of information and data are included in parentheses after main text/numbers. It is not always clear what the definition of all of these are. It would be helpful to the reader to give these definitions in footnotes. To simplify this, it might be useful to use different styles of parentheses e.g. (, {, [to distinguish where different types of data are in parentheses in different lines of the table. While subsequent tables are internally more consistent, it might be helpful to extend this principal of explicit footnoted definitions to all tables.
--	---

REVIEWER	Elisabete Pereira Silva Universidade Federal de Pernambuco, Materno-Infantil
REVIEW RETURNED	27-Jan-2021

GENERAL COMMENTS	The article addresses an important topic for understanding the interface between children's exposure to different subtypes of IPV and negative consequences for child development. The article is written in clear and correct language. However, I will highlight some points that need to be clarified by the authors. ABSTRACT It is well structured and written in a concise and easy-to-read form, highlighting the main points of the article. INTRODUCTION It presents the study problem in a concise and well-structured manner, placing the significance of the study on the basis of relevant literature and defining the objectives of the study. However, more recent articles could be cited and some inconsistencies need to be corrected. Page 5 – 1st paragraph: The sentence attributed to reference 1 needs to be revised because, to my knowledge, this reference cites physical and sexual violence. Does not speak of emotional violence, cites data on controlling behavior: “The World Health Organization’s multicountry study on violence against women found high lifetime prevalence of physical IPV (up to 61%), sexual IPV (up to 59%) and emotional IPV (up to 75%).”¹ Page 6 – 1st sentence of the 1st paragraph: The sentence attributed to references 16-19 needs to be revised because these references do not study child development, they focus on IPV in women: “Research focusing on child development and IPV often narrowly defines IPV as only physical or sexual
--

abuse, though emotional IPV has been linked to similar or increased rates of ill-health, both physical and mental, compared to physical IPV.¹⁶⁻¹⁹

Page 6 – 2nd sentence of the 1st paragraph:
“Additionally, the majority of studies, particularly in school-aged children, have been done in high income countries (HICs), though LMICs have a higher burden of IPV associated risk factors, such as community violence, trauma, substance use disorders and mental illness and limited support networks.” What are these studies? References are missing.

METHODS

They were described in a detailed and clear way. An important point is that the authors made imputation missing data.

Page 9 – Topic *Intimate partner violence*: The following sentence needs to be revised: *“...inventory adapted from the WHO multicountry study³⁸ and the Women’s Health Study in Zimbabwe³⁹.”* Because the reference 38 is not from the WHO Multicountry Study and the reference 39 used a questionnaire already adapted from the WHO Multicountry Study. I suggest using an original reference of the WHO Multicountry Study.

RESULTS

The results are well organized and well presented in tables and figures. However, there is an inconsistency in the sample size in table 2. I suggest checking and put the missing values.

DISCUSSION

It is well structured, dialogues the results with the previously published literature. It presents the potential limitations of the study, but it presents implications and consistent conclusions to facilitate future research and clinical practice.

REFERENCES

They can be updated. Of the 59 references, only 28 are from 2010 to 2020.

VERSION 1 – AUTHOR RESPONSE

Reviewer 1 comments:

1. Abstract; Page 4, line 33; The statement "Interventions to reduce maternal IPV, including emotional IPV,... are needed" does not accurately reflect the findings in these populations as the research results focus only on Emotional and physical IPV. Suggest splitting this into a motherhood and apple pie statement about the desirability of intervening to reduce all forms of IPV and then highlighting that emotional and physical IPV interventions are needed in the populations to protect child development.
Thank you, the phrasing in this sentence has been edited to focus on the need for interventions for both physical and emotional IPV to promote child development.
2. Page 6, line 22; It would be helpful to clarify that the roots of violence are as much affected by the ongoing impact of past colonialism and apartheid on society - not just what they did to society in the past.
This has been clarified to make the point that South Africa's colonial and apartheid past continue to impact violence in the country.
3. Page 7, line 57; It would be helpful to describe here more explicitly what information was obtained longitudinally i.e. during pregnancy and at birth of the child and what was obtained cross sectionally i.e. at two years since the birth.
This is included in the footnote of table 1 and in the description of data collection within methods, however, a note has been added here, to line 57 to note that only birthweight and HIV exposure were collected at birth and all other data at 2 years.
4. Page 25, table 1; A range of different types of information and data are included in parentheses after main text/numbers. It is not always clear what the definition of all of these are. It would be helpful to the reader to give these definitions in footnotes. To simplify this, it might be useful to use different styles of parentheses e.g. (, {, [to distinguish where different types of data are in parentheses in different lines of the table.
To ensure clarity, those variables that weren't explicitly labelled have now been labelled. This only applies to frequencies (percent) but has been added so that now all variables presented in table 1 are labelled.

Reviewer 2 comments:

1. Page 5 – 1st paragraph: The sentence attributed to reference 1 needs to be revised because, to my knowledge, this reference cites physical and sexual violence. Does not speak of emotional violence, cites data on controlling behavior: "The World Health Organization's multicountry study on violence against women found high lifetime prevalence of physical IPV (up to 61%), sexual IPV (up to 59%) and emotional IPV (up to 75%).¹"
Thank you, the WHO multicountry study does focus on physical and sexual violence but also cites prevalence of emotional abuse as well as controlling behaviours. Emotional abuse was found in 75% of women (inclusive of being insulted, humiliated or belittled, which comprise emotional IPV); controlling behaviours were found as high as 90%, but this statistic has not been included in the manuscript. We have changed this reference to refer to the full WHO report rather than the summary article in Lancet.
2. Page 6 – 1st sentence of the 1st paragraph: The sentence attributed to references 16-19 needs to be revised because these references do not study child development, they focus on IPV in women: "Research focusing on child development and IPV often narrowly defines IPV as only physical or sexual abuse, though emotional IPV has been linked to similar or increased rates of ill-health, both physical and mental, compared to physical IPV.¹⁶⁻¹⁹"
Thank you, these references are to support the statement that "emotional IPV has been linked to similar or increased rates of ill-health, both physical and mental, compared to physical IPV"; for clarity, we have added references to support the first part of this sentence, which focus on IPV and child development.
3. Page 6 – 2nd sentence of the 1st paragraph: "Additionally, the majority of studies, particularly in school-aged children, have been done in high income countries (HICs), though LMICs have a higher burden of IPV associated risk factors, such as community violence, trauma, substance use disorders and mental illness and limited support networks." What are these studies? References are missing.
Thank you, references for this have been added.

4. Page 9 – Topic Intimate partner violence: The following sentence needs to be revised: “...inventory adapted from the WHO multicountry study³⁸ and the Women’s Health Study in Zimbabwe³⁹.” Because the reference 38 is not from the WHO Multicountry Study and the reference 39 used a questionnaire already adapted from the WHO Multicountry Study. I suggest using an original reference of the WHO Multicountry Study.
Thank you for pointing this out – reference 38 has been removed and we now include a reference to the original WHO multicountry study.
5. Results: However, there is an inconsistency in the sample size in table 2. I suggest checking and put the missing values.
Thank you for pointing this out, we have check values throughout tables and clarified where inconsistencies were found, specifically in table 1 and 2 and have included an indication of where data were missing for some variables.
6. References: They can be updated. Of the 59 references, only 28 are from 2010 to 2020.
Thank you, we have updated multiple references (highlighted in manuscript) to ensure these are more current where possible. Many of the older references are linking seminal studies or original measure instrument validations or scoring instructions.